# Non-cognitive skills and labour market performance of immigrants

**Alpaslan Akay** [1,3,4] *, **Levent Yilmaz** [2]

**1** Department of Economics, University of Gothenburg, Gothenburg, Sweden, **2** Department of Economics, Turkish-German University, Istanbul, Türkiye, **3** Department of Economics, Universidad Antonio de Nebrija, Madrid, Spain, **4** IZA, Bonn, Germany

☉ These authors contributed equally to this work.
* alpaslan.akay@economics.gu.se

**Data Availability Statement:** Our individual dataset is the German Socio-Economic Panel dataset (SOEP), which includes long panels of natives and immigrants observed between 1984 and 2013. The annual panel survey was first launched in 1984 in West Germany among 12,000 households, which have been followed since then. The sample has

## Abstract

This paper investigates how non-cognitive skills relate to the relative labour market performance of immigrants. Using the German Socio-Economic Panel (SOEP) and the Five-Factor Model of personality as a proxy for the non-cognitive skills, we show that these skills matter for the labour market integration of immigrants in the host country. We use two comparison benchmarks. Compared to an average native, immigrants' non-cognitive skills, e.g., extroversion or emotional stability, can lead to 5–15 percentage points lower lifetime employment probability disadvantage implying a better overall integration. Comparing immigrants and natives with the same type and level of non-cognitive skills suggests that returns of extroversion and openness to experience are higher among immigrants, leading to 3–5 percentage points lower lifetime employment probability disadvantage. These results are robust with respect to self-selection, non-random returns to the home country, stability of personality, and estimators. Our detailed analysis suggests that non-cognitive skills (especially extroversion) are substitutes for the standard human capital measures (e.g., formal education and training) among low educated immigrants, while there is no significant relative return of non-cognitive skills among highly educated immigrants.

## Introduction

The standard model of economic integration suggests that immigrants' performance is a function of host-country specific skills obtained through formal education in schools and retraining, language acquisition, learning local culture, and getting information about the local labour market in the host country (e.g., [1–3]). Immigrants arrive with a particular level of human capital and then develop host-country specific skills over time, which can make their labour market outcomes (e.g., employment probabilities or wages) eventually reach those of comparable natives (e.g., [4–7]). This integration framework focuses mainly on the standard human capital measures to capture skills, and ignores the role of "non-cognitive" aspects of human capital on the immigrants' relative labour market performance. Yet, one intriguing question is whether

been extended over the years, most notably by including about 2,000 East German households in 1990. The dataset is rich in terms of socio-demographic and -economic characteristics of natives and immigrants as well as regional economic information, e.g., local unemployment rates. The collecting institution is DIW, Berlin, Germany. The information about the dataset can be found in the following link: https://www.diw.de/en/diw_01.c.678568.en/research_data_center_soep.html SOEP data are personal data that are subject to special protections in Europe. Because of this, there are special conditions for using the data (see www.diw.de). The dataset can be obtained upon a request and filling a permission document (the contact person is Jan Goebel). By signing a data distribution contract, you agree to comply with these conditions. We have signed contract with DIW to use the dataset. The contract can be obtained by contacting Jan Goebel. The SOEP-Core dataset contains all federal states and a distinction between urban and rural areas. We use INKAR data (freely available in www.inkar.de) to get regional information (Raumordnungsregionen, ROR). To merge SOEP-Core and regional data, researcher sign further contracts with DIW as explained above. You can contact with the SOEPhotline via phone: +49 30 89789 - 292 or email: soepmail@diw.de or get information about data access under the link: https://www.diw.de/en/diw_01.c.601584.en/data_access.html#c_741353.

**Funding:** We appreciate the financial support from the University of Gothenburg". The funders had no role in study design, data collection and analysis, decision to publish, or preparation of the manuscript. All funding sources are mentioned withing the paper.

**Competing interests:** The authors have declared that no competing interests exist.

immigrants' innate or intrinsic characteristics, for instance, "tendency to learn the new culture faster" or "tendency to be social", help them to have a better economic performance in the host country. Indeed, research in social psychology and more recently in mainstream economics has already yielded promising evidence that these innate characteristics of individuals, i.e., proxies of non-cognitive skills, including imagination, social skills, and openness to new cultures among many others, might play important roles in a wide range of individual economic outcomes (e.g., [8–12]). To the best of our knowledge, this paper is among the first to explore whether and how non-cognitive skills help *first-generation* immigrants in their integration process and how standard human capital measures (e.g., formal education) and non-cognitive skills interact as a determinant of immigrants' labour market success in the host country.

The focus of our analysis is on the employment integration of immigrants over the life-cycle as a function of their non-cognitive skills. To investigate this, we first bring together two strands of the literature: the return of non-cognitive skills on the employment outcomes of individuals (e.g., [9–12]) and immigrants' labour market integration in a host country (e.g., [2, 6, 7, 13]). To this end, we exploit a rich sample of immigrants in Germany. The integration of immigrants has been an important item on the political agenda for at least the past 20 years in Germany, where 17% of the population has an immigration background (e.g., [14]). Previous research has largely demonstrated that both first- and second-generation immigrants have performed relatively poorly in the German labour market compared to similar natives [14, 15]. A large part of the literature relating to immigrants' integration in Germany focuses on the relative wages of immigrants (e.g., [16–19]). These studies report that immigrants' wages do not catch up with those of natives. Bartolucci [20] finds a 13% wage difference between immigrants and natives using an employer-employee matching dataset (see also [21]). A limited literature also finds that immigrants' employment probabilities are lower than comparable natives and they experience higher state dependence on their unemployment (e.g., [22]). Thus, it is crucial to unveil the reasons of the poor economic performance of immigrants by enhancing the standard integration model. We contribute to these two kinds of literature by investigating how employment probability integration of immigrants relates to their non-cognitive skills and whether these skills explain the lifetime employment probability differentials between immigrants and natives in Germany.

One of the key issues to this study is to proxy non-cognitive skills. We rely on measurements derived from the Five-Factor Model (Big-5 hereafter) [23–25], a widely accepted instrument to measure personality at the broadest level and often applied in the empirical literature as a measure of non-cognitive skills (see [10, 11] for reviews). These characteristics are summarised under the titles of *extroversion*, *emotional stability* (reversed neuroticism), *conscientiousness*, *openness to experience*, and *agreeableness*. The recent literature has already accumulated a plethora of information on how outcomes of individuals relate to these five major dimensions of innate characteristics. These outcomes include employment, productivity, earnings, and wages (e.g., [8, 26–31]), life satisfaction, and social preferences (e.g., [10, 12, 32]). In this paper, we focus on the employment outcomes of individuals. For instance, people who score high on most of these characteristics participate more, have longer working hours with shorter unemployment durations (e.g., [30]), experience lower probability of unemployment (e.g., [30, 33–35]), lower state dependence in unemployment (e.g., [36]), less frequent workplace deviance (e.g., [37]), stronger leadership and career success (e.g., [38]), higher job performance and satisfaction (e.g., [35, 39–41]). This brief summary includes papers using only the Big-5 personality characteristics as a measure of non-cognitive skills. Another group of studies uses "locus of control" and "reciprocity norms" to relate these aspects of non-cognitive skills with the employment outcomes of individuals (see [9, 42, 43]). For instance, Heckman and the others [9] find that early measures of "self-esteem" and locus of control have a significant effect on employment status and wages.

Our empirical analysis uses a long panel of immigrants from 1984 to 2013 and three waves of Big-5 measures collected in Germany (German Socio-Economic-Panel—SOEP). The econometric strategy is based on estimating employment probabilities as a function of years of education, year since migration, arrival cohorts, and observation periods, conditional on all other immigrants' and natives' individual, household, and regional characteristics. Then, the model specification explores the heterogeneity in the economic integration of immigrants with respect to non-cognitive skills. Using the standard economic 'integration model' (e.g., [2]), we first explore the benchmark employment probability integration levels of first-generation immigrants in Germany. Indeed, the result suggests a smooth textbook-fashion integration process of immigrants in Germany. Immigrants' initial (upon arrival or entry to the labour market) employment probability disadvantage in Germany is about −27.4 percentage points, on average. Immigrants quickly close this differential during the first years of immigration. Yet, the integration process becomes steady and slower later: immigrants fully catch up with the employment probability of comparable natives about 25 to 30 years after arrival. Overall, the mean *lifetime* employment probability differential (*MLD*, "mean lifetime differential" hereafter) estimated through 45 years of immigration experience of immigrants in Germany is about −9.6 percentage points, on average.

We then augment the model by introducing measures of non-cognitive skills in an interaction with the time spent in the host country. We use two comparison benchmarks. In the first scenario, immigrants with low and high non-cognitive skills are compared with an average native. The second scenario compares immigrants and natives with the same type and level of non-cognitive skills. There is a significant role of non-cognitive skills in the integration process in both comparisons. In the first scenario, immigrants who score *high* on extroversion (i.e., higher social skills and frequent social interactions), emotional stability (i.e., less depression and high life satisfaction), conscientiousness (i.e., hard work, self-discipline, and higher competition), and openness to experience (i.e, valuing arts, aesthetics, and creativity) obtain about a 5–15 percentage points lower *MLD* compared to immigrants with a *lower* level of these skills. This result implies that immigrants who score high on these skills fully integrate into the labour market performance of an average native. Comparing immigrants and natives with the same type and level of non-cognitive skills suggests that only extroversion and, to some extent, openness to experience generate significant differences in the *MLD* of immigrants. Immigrants who score high on these non-cognitive skills integrate better and experience 3–5 percentage points lower *MLD*. Finally, the results among low- and high-educated immigrants and natives unveil that the non-cognitive skills (especially extroversion) are *substitutes* for the formal education, as they only operate among immigrants with *lower* years of education. The results are also highly heterogeneous with respect to gender (driven by females) and country of origin of immigrants (driven by non-EU immigrants and guest workers) and robust to self-selection, non-random returns of immigrants, stability of non-cognitive skills over time, and estimators.

The remainder of the paper is organised as follows: the next section presents the framework, potential channels, and predictions for each non-cognitive skill. Section 3 gives the data, measurements, and descriptive statistics. Section 4 presents the empirical strategy, identification, and measures for the lifetime levels of integration. Section 5 gives the main results, heterogeneity analysis, robustness analysis, and an investigation of whether skills obtained through formal education and non-cognitive skills are substitutes or complementary to each other. Finally, Section 6 concludes.

## Non-cognitive skills, employment, and integration

### Framework and potential channels

Our economic framework is based on a human capital model enhanced by non-cognitive skills (e.g, [1]). The conceptual model assumes that the employment outcomes ($d$) of immigrants $M$ and natives $N$ are affected by the standard measures of human capital (e.g., formal education, training, speaking local language) $C$ and non-cognitive skills $NC_p$ during a specific age $AGE$ and year since migration $YSM$. Immigrants arrive with a particular skill endowment, which they use effectively as the year since migration increases: $d^M = f^M(AGE^M, YSM, C^M, C^M \times YSM, NC_p^M, NC_p^M \times YSM)$, where $p$ is a non-cognitive skill type, measured with Big-5 personality characteristics. The corresponding model for the natives is $d^N = f^N(AGE^N, C^N, NC_p^N)$. The gap in economic outputs between comparable immigrants and natives $d^M - d^N$ is reduced over time as the immigrants learn host country-specific skills and efficiently use their non-cognitive skill endowment.

From the decision-makers' perspective, there are two policy-relevant comparison benchmarks. The first is to compare the employment outcomes of immigrants with a particular level of non-cognitive skill to the outcomes of an *average native*. The second is to compare the outcomes among immigrants and natives with *the same type and level of each non-cognitive skill*. For the first comparison, the aim is to obtain the *absolute return* of having a particular level of non-cognitive skill, while the aim with the second comparison is to measure whether there is a *differential or relative return* of non-cognitive skills for immigrants in comparison to a native with the same type and level of non-cognitive skills. In this study, we conduct an analysis for both comparisons as they involve different sets of assumptions about how immigrants are self-selected for immigration in and out of the host country and how immigrants and natives are sorted by their intrinsic characteristics in the labour market.

Why do non-cognitive skills relate to the relative employment outcomes of immigrants? As twin studies and cross-cultural analysis show (e.g., [44, 45]), Big-5 personality characteristics are universal, and they are expected to affect everyone in the population, including both immigrants and natives. The returns of non-cognitive skills on the employment probability of immigrants and natives are also expected to be the same, and there should be no differential effect on the relative performance of immigrants on average. The heterogeneity in relative employment probabilities of immigrants by their non-cognitive skills in comparison to an average native, i.e., absolute return, can be explained with the overall role of these skills in the wide range of labour market outcomes (e.g., productivity, job satisfaction, carrier choice, job search, among others), which are investigated by several studies (e.g., [10, 30, 33, 35]). Yet, any difference in the outcomes between immigrants and natives with the same non-cognitive skill level, i.e., relative return, can be further explained by channels relating to the host country's labour market-related irregularities, including the sentiments of native employers, labour market discrimination towards immigrants by non-cognitive skills, and the inability of some immigrants, as well as natives, to operationalise their non-cognitive skills in the labour market. Immigrants who are better able to signal their intrinsic characteristics might be sorted by the native employers to find a job faster and develop higher levels of host country-specific labour market skills over time. This might be the case, for instance, when native employers positively read signals relating to a higher level of sociability and open attitudes to local culture; such employers might discriminate against less social immigrants or they might consider them as less productive workers compared to similar natives. In these cases, the relative return to non-cognitive skills might differ for immigrants and natives, generating different patterns of employment probability integration process.

To be able to combine non-cognitive skills into the standard integration framework, a set of compelling assumptions is required. As in most studies, we assume that the non-cognitive skills are time-invariant, at least they change very slowly over time, and they are only partially influenced by major life events, e.g., divorce (e.g., [46–48]). Indeed, there is a large collection of studies showing that Big-5 personality characteristics are formed before age 20 and stay relatively stable during the life-cycle [23, 47, 49–51]. We cannot also rule out the potential non-random sorting of immigrants in and out of the host country by their non-cognitive skills (e.g., [52, 53]). To investigate the role of these alternative explanations, we focus on first-generation immigrants who arrived in the host country when they were young, before they could have made the immigration decision on their own. We then focus on the role of non-random return of immigrants by their non-cognitive skills. Additionally, immigrants might not be able to productively transfer their non-cognitive skills upon entry into the market due to a lack of knowledge about the local language, culture, and local labour market-specific knowledge (e.g., [54]). Throughout the immigration process, with the development of host country-specific human capital, immigrants' non-cognitive skills are expected to kick in and be productively used in their integration process. We extensively investigate these potential explanations in our robustness analysis.

## Predictions

Using the previous literature on personality psychology and the research in economics exploiting the non-cognitive skills, we can make a *priori* predictions about the potential effect of each skill type on the degree of employment probability integration. A higher score of *extroversion* among immigrants is expected to be associated with a higher frequency of social interactions, ambition, and desire to learn the local culture [11]. These immigrants might learn about job possibilities through their social interactions, which might correlate with a higher probability of employment [10, 11, 55, 56]. However, these immigrants might not be able to signal their level of extroversion until they learn the local culture and language, for instance. Thus, we expect that the positive effect of extroversion on the level of integration would gradually increase over time. Overall, we hypothesise that the level of integration would be higher among immigrants who score high on extroversion. Another type of skill that we expect to have a positive effect on the level of employment probability integration is the *emotional stability*. This skill type is obtained by reversing *neuroticism*, which is associated with negative emotions, depression, and lower levels of life satisfaction. A higher score on emotional stability is expected to be positively correlated with employment probability, hours of work, productivity, and job search efficacy [57, 58]. Therefore, an immigrant who scores high on emotional stability is also expected to obtain better employment integration outcomes.

*Conscientiousness* is directly related to work-related domains of life. It is found to be strongly related to hard work, work ethics, self-discipline, competition, and a higher degree of goal-oriented behaviour (e.g., [10, 11]). Thus, we predict that immigrants with higher conscientiousness integrate better. The literature also suggests that people who score high on *openness to experience* might experience a positive boost in their employability. This type of skill implies that individuals value arts, aesthetics, and give importance to intelligence and creativity (e.g., [11]). Immigrants who score high on this measure might integrate into the local culture faster, learn languages, and get better information about the local labour market. Thus, immigrants scoring high on this skill type are expected to find a job more quickly, communicate with coworkers better, and catch up faster with the employment level of natives.

Finally, a high score on *agreeableness* is associated with cooperative behaviour, altruism, and a lesser desire for conflict. Previous research suggests that these individuals are more

successful in the labour market (e.g., [58]). Yet, one critical remark is that the labour market conditions that immigrants and natives face might be different. In particular, immigrants might experience higher labour market competition compared to similar natives. Even though immigrants who score high on agreeableness are expected to be highly motivated for work, work longer hours with no complaint and display less workplace deviation, they also tend to avoid conflict with others, which can render immigrants unsuccessful in competition with natives (e.g., [37, 59]). Therefore, the direction of its effect on the employment probability integration is a priori unknown.

## Data

### Sample selection

Our individual dataset is the German Socio-Economic Panel dataset (SOEP), which includes long panels of natives and immigrants observed between 1984 and 2013. (see www.diw.de and https://paneldata.org, for further information about the dataset). The annual panel survey was first launched in 1984 in West Germany among 12,000 households, which have been followed since then. The sample has been extended over the years, most notably by including about 2,000 East German households in 1990. The dataset is rich in terms of socio-demographic and -economic characteristics of natives and immigrants as well as regional economic information, e.g., local unemployment rates. We obtain several individual characteristics to be used in the regression analysis below. The full set of those characteristics is given in Table A2 in S1 File; they include age, year since migration, gender (female = 1), marital status (four dummies as married, single, divorced, and widowed), years of education, health status (five dummies as very bad, bad, neither bad nor good, good, and very good), household size, number of kids, capital income (income from rents, interest, and dividends), refugee status (=1), remittances sent to the people back home, and dummies for whether spouse (=1) and kids (=1) live in the home country.

The sample selection is the same for the immigrants and natives. We include all individuals in SOEP aged between 18 and 65 to prevent age and retirement-related confounders. All immigrant countries of origin are used in the analysis (see Table A3 in S1 File for the list of top immigrant countries in the dataset). The analysis focuses on first-generation immigrants. They are defined as the people born outside of Germany, with or without German citizenship through naturalisation. Thus, the analysis excludes the generations of immigrants born in Germany. However, some first-generation immigrants arrive in Germany at an early age. In our analysis below, we include all immigrants and report analysis for alternative groups of immigrants who have arrived in Germany at different ages. Some of the key variables involve missing values, and thus we exclude those immigrants and natives from the analysis. Having cleaned all missing values, the total number of observations for immigrants is 28,582 immigrant-year observations, while the total number of native-year observations is 208,064 observed between 1984 and 2013. One of the key issues of this study is whether there is a differential non-random attrition of immigrants and natives in the dataset. The SOEP is also shown to have low attrition [60], and importantly, the effect of Big-5 personality types on panel attrition is found to be either non-significant or comparably small in SOEP data (e.g., see [52]).

### Measures

**Outcome of interest.** The integration outcome that we investigate is the employment status of immigrants. The measure captures the current employment activity as a dummy variable. It takes a value of one when individuals report a positive number of working hours (active people) for each wave of observation and otherwise zero. We eliminate the self-employed because the labour market conditions and non-cognitive skills required to become

self-employed substantially differ from those working as paid employees [58]. We also exclude individuals who are out of the labour market and keep only those non-employed people who are actively seeking a job. The statistics about the employment status are given in Table A2 in S1 File. The mean share of active people among natives is 75.2% while it is 68.1% among immigrants (the differences are highly statistically significant). While the average employment rate among the EU immigrants is highly similar to that of natives, the raw lifetime employment rate differential (i.e., raw *MLD*) between non-EU immigrants (mostly guest workers) and natives is more than −10 percentage points.

**Non-cognitive skills: Five-factor model.** Our measure of non-cognitive skills is based on the Big-5 personality traits [24]. The inventory of Big-5 is presented in Table A1 in S1 File, which includes 15 statements. This short version of the inventory is reliable and sufficiently captures the dimensions of Big-5 personality domains [12, 24]. The responses to each statement in the inventory are obtained for immigrants and natives using a seven-point ordinal scale from 1 "*it does not apply at all*" to 7 "*it applies fully*". As mentioned above, the Big-5 measures personality in five domains labelled as extroversion, emotional stability (or neuroticism when reversed), conscientiousness, openness to experience, and agreeableness. To calculate the measures, each domain uses the scores of three statements, and thus the final scales are measured from 3 to 21. The statements used to determine the measure of each personality domain are given in the note for Table A1 in S1 File.

The SOEP dataset includes measures of Big-5 personality characteristics only for three waves: 2005, 2009, and 2013. There are alternative ways to make use of these data. Following the literature, we first assume that non-cognitive skills are stable after adolescence. We define a time-invariant measure for each Big-5 personality type by averaging three scores obtained during eight years from 2005 to 2013. When there are missing values in any of these waves, we calculate the average measure using the remaining non-missing waves. In our analysis below, we conduct a sensitivity check where we use only waves 2005 and 2013 to investigate the sensitivity of our results. The descriptive statistics of each measure split by immigrants and natives are given in Table A2 in S1 File. The basic statistics of each personality measure are highly similar among immigrants and natives for conscientiousness and agreeableness. The mean values of extroversion, emotional stability, and openness to experience are slightly lower among immigrants. Yet, the differences are not statistically significant at the conventional significance levels. Overall, there is a substantial congruence between the personality measures of immigrants and natives except for some groups of immigrants.

Another critical issue is that the Big-5 personality characteristics of immigrants might be a function of the year since migration, which is the key measure to investigate integration. To obtain some descriptive evidence, we calculate the mean level of each personality type as a function of the year since migration. Indeed, patterns presented in Fig A1 in S1 File suggest that the Big-5 personality characteristics are highly stable over the duration of stay between 0 and 45 years. Another descriptive analysis that we report compares the kernel density estimates of each personality characteristic among immigrants and natives. These results are presented in Fig A2 in S1 File, where we also split the distribution of each personality type among immigrants who stay in the host country for less than the median stay of 20 years (orange plots) and more than or equal to the median (blue plots). The distributions are highly similar between immigrants and natives (dotted black plots), and there are no substantial differences between the distributions as a function of the year since migration.

**Statistics.** Finally, we present basic statistics on the employment status among immigrants and natives with a low and high levels of each non-cognitive skill type. Fig 1 presents mean employment probability levels in each bar. Natives are shown with blue bars, while immigrants are shown with green bars. The solid red horizontal line shows the overall average employment

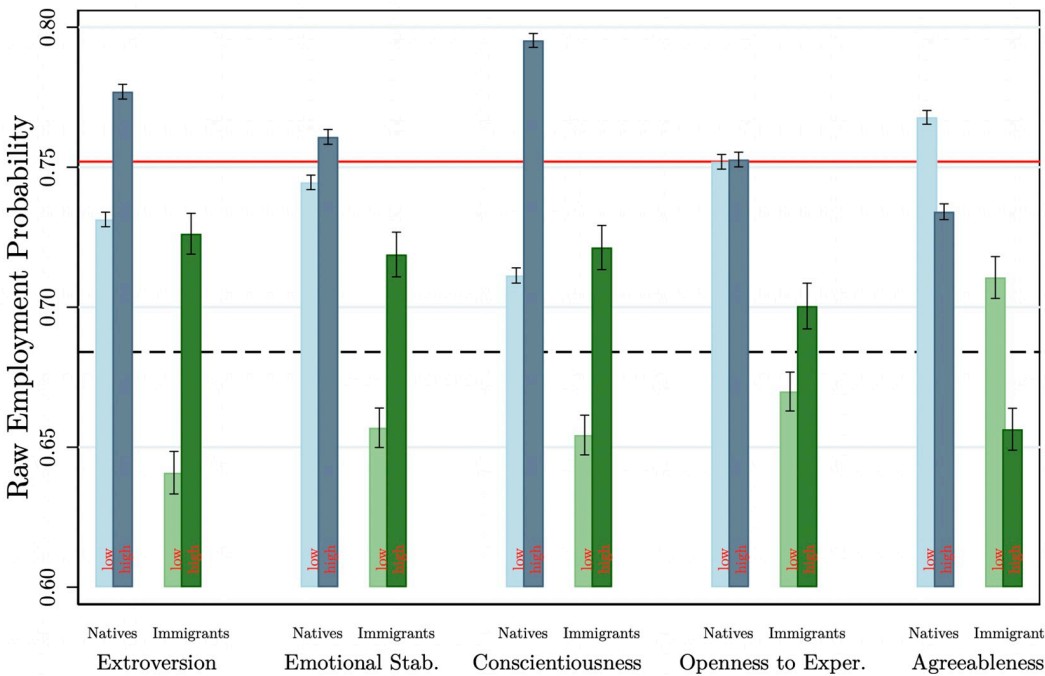

**Fig 1. Descriptive results: Personality types and raw employment probabilities.** Authors' own calculations from the SOEP (1984–2013). Bars show the raw share of employment of immigrants and natives across the types of Big-5 personality characteristics. The high and low level of personality is defined using the median of each personality measure as the threshold. The solid red line represents the mean employment probability level among the natives while the dashed black line gives the mean employment probability among the immigrants. The confidence intervals are given on the tip of each bar with the vertical lines.

rate of natives (0.752), while the dashed black line is the average employment rate of immigrants (0.681). The light blue and light green bars show the mean employment rate for the lower levels (less than the respective median) of each non-cognitive skill type, and the dark blue and dark green bars show the mean employment rate for the higher levels of each skill type. There are clear patterns. Immigrants and natives who score high on extroversion, emotional stability, conscientiousness, and openness to experience have higher employment rates compared to immigrants and natives having lower levels of these characteristics. It is reversed for agreeableness. A higher level of each skill type (except agreeableness) relates to a higher employment rate among immigrants, which is also closer to an average native (red line). Comparing immigrants and natives with the same type and level of each skill unveils additional patterns (compare dark blue and dark green bars). For instance, immigrants who score high on extroversion, emotional stability, and openness to experience achieve better employment status integration outcomes than those with lower levels of these skills.

## Econometric specifications

### Modelling employment integration

The modelling strategy is based on the framework used by several papers in the immigration literature (e.g., [2, 3]). The approach estimates the employment status on the years of education, year since migration, age, arrival cohort effects, the observation period effects, and other immigrant and native specific characteristics. We modify this framework by allowing

specifications for the non-cognitive skills in interaction with the years of education, year since migration, and age. The generic employment probability generating process for immigrants $d_{it}^M$ is defined as

$$
d_{it}^M = 1(X'^M \beta^M + \phi_1^M AGE_{it}^M + \phi_2^M AGE_{it}^{2,M} + \psi_1^M YSM_{it} + \psi_2^M YSM_{it}^2
$$

$$
+\theta^M C^M + \sum_{p=1}^{5} \pi_p^M NC_p^M + \sum_{p=1}^{5} \gamma_p^M C^M NC_p^M
$$

$$
+\sum_{p=1}^{5} NC_p^M (\phi_{1,p}^M AGE_{it}^M + \phi_{2,p}^M AGE_{it}^{2,M} + \psi_{1,p}^M YSM_{it} + \psi_{2,p}^M YSM_{it}^2) \tag{1}
$$

$$
+AC_k + \Pi_t^M + H_c + \rho_r^M + \alpha_i^M + \varepsilon_{it}^M > 0),
$$

and for natives

$$
d_{it}^N = 1(X'^N \beta^N + \phi_1^N AGE_{it}^N + \phi_2^N AGE_{it}^{2,N}
$$

$$
\theta^N C^N + \sum_{p=1}^{5} \pi_p^N NC_p^N + \sum_{p=1}^{5} \gamma_p^N C^N NC_p^N + \sum_{p=1}^{5} NC_p^N (\phi_{1,p}^N AGE_{it}^N + \phi_{2,p}^N AGE_{it}^{2,N}) \tag{2}
$$

$$
+\Pi_t^N + \rho_r^N + \alpha_i^N + \varepsilon_{it}^N > 0).
$$

In these model specifications, $d_{it}^M$ and $d_{it}^N$ are two dummy dependent variables indicating employment status of immigrants and natives observed at year $t = 1984, \ldots, 2013$ for each individual $i$. There are common and different characteristics in the employment generating functions of immigrants and natives. $X$ is the matrix of observable characteristics, commonly including gender, marital status, health status, non-labour or capital income, household size, and the number of kids, among others (see Table A1 in S1 File for the full set of control variables and descriptive statistics). $\beta^N$ and $\beta^M$ are two vectors of parameters corresponding to these control variables.

Two key characteristics to investigate the employment probability integration of immigrants are the age *AGE* and year since migration *YSM*. The model specification for immigrants includes the quadratic function of age, $AGE^2$, and year since migration, $YSM^2$. $\phi_1^M, \phi_2^M, \psi_1^M$, $\psi_2^M, \phi_1^N$, and $\phi_2^N$ are the corresponding parameters to be estimated. The model specifications include the labour market-specific skills of immigrants $C^M$ and natives $C^N$, which are measured using formal years of education. $\theta^M$ and $\theta^N$ are the corresponding parameters to be estimated. The specifications include five non-cognitive skill types of immigrants $NC_p^M$ and natives $NC_p^N$, where $p$ stands for extroversion, emotional stability (or neuroticism when reversed), conscientiousness, openness to experience, and agreeableness. $\pi_p^M$ and $\pi_p^N$ are the corresponding vectors of parameters for immigrants and natives. We then specify a rich and flexible functional form involving the interaction between each non-cognitive skill type and years of education, age, age-squared, year since migration, and year since migration squared to be able to capture the life-cycle relationship between non-cognitive skills and relative employment probability of immigrants.

## Stochastic specifications, identification, and estimators

**Error specifications.** The error specifications in Eqs (1) and (2) include a rich set of unobserved components. Key components for our model of integration are the *arrival cohort* effects $AC_k$ and the observation *period* effects $\Pi_t^M$. While the arrival cohort effect aims to capture economic and political conditions prevailing when immigrants arrive in the host country, the

period effects capture the global shocks on the overall economy of the host county during the year of observation. The macroeconomic conditions during the year in which the immigrant arrives might affect future performance. The conditions during the year in which immigrants are observed (e.g., due to shocks in the labour market, changing sentiments and attitudes towards immigrants) might also differentially affect natives and immigrants.

Age, year since migration, arrival cohort, and period effects are not identified in the same specification due to their mechanical correlation. To deal with this identification issue and allow measures for all in the same specification, we define grouped dummies for the arrival cohort effects indicating five-year periods from 1965 to 2013. We consider year dummies for each year from 1984 to 2013 as proxies for the period effects. To deal with the period effects further, we also allow Eqs (1) and (2) for the *local market characteristics* at the period of observation [61]. The local market regions are defined using 96 ROR (*Raumordnungsregionen*) classifications (see www.inkar.de for the local market data). These characteristics include the local unemployment rate among people, unemployment rates among foreigners, and the local GDP per capita. The data generating process of immigrants (Eq 1) is also conditioned on the country of origin dummies $H_c$. There are several countries of origin in the data. However, some of them have a very low number of observations. We define dummies for the top 23 countries of origin, and the rest is organised as a dummy for 'others' (see Table A3 in S1 File for the descriptive statistics by the country of origin). To capture regional heterogeneity further, we use 16 Federal State dummies (*Länder*) $\rho_r^M$. The term $\alpha_i^M$ stands for the time-invariant individual effect, and finally, $\varepsilon_{it}^M$ is the usual error term. The corresponding terms in the error specification of natives (Eq 2) are defined in the same way.

**Estimators and unobserved heterogeneity.**   It is crucial to allow for the time-invariant unobserved individual characteristics $\alpha_i^M$ and $\alpha_i^N$ in our analysis as these characteristics capture other trait-like individual characteristics (e.g., risk-taking behaviour, genetic predisposition). Ideally, the model specifications should exploit the panel dimension with the fixed-effects in which the non-cognitive skills and the time-invariant heterogeneity are assumed to be correlated. There are two crucial econometric problems. The first is that the parameters of year since migration, age, and time-dummies (i.e., measures for period effects) cannot be identified together within the same fixed-effects specification. The second is that the specifications involve dummy dependent variables, and the fixed-effects model specification with "brute-force" is biased in non-linear discrete choice models due to the so-called incidental parameters problem. Instead, we use a random-effects estimator in which we allow for the *within means* of time-variant variables to capture the correlation between observed and unobserved characteristics, i.e., a correlated random-effects model [62]. We also note that estimating random-effects discrete choice models is highly time-consuming, especially when we calculate the adjusted predictions and their standard errors for several points of age, year since migration, and non-cognitive skills. We use the correlated random-effects linear probability model specification to make our extensive empirical analysis tractable. The time-variant variables used in the auxiliary distribution of the correlated effects are the household size, number of kids, remittances, non-labour income, health status, and years of education. We also estimate a pooled probit model to check for the robustness of our results with respect to the choice of model specifications and estimators.

**Measuring integration.**   We formulate a consistent measure for the degree of integration to efficiently demonstrate how the level of integration varies by the levels of non-cognitive skills. The main aim is to predict the employment probability differential of immigrants at a particular age, year since migration, and non-cognitive skill level while holding all other variables at their mean levels for immigrants and natives. Thus, the *relative employment probability*

$REP$ at a particular $AGE$, $YSM$, $NC_p^M$, and $NC_p^N$ is defined as

$$REP(AGE, YSM, NC_p^M, NC_p^N) = E[d^M|AGE, YSM, NC_p^M|\overline{\mathbf{X}}^M]$$
$$-E[d^N|AGE, NC_p^N|\overline{\mathbf{X}}^N]. \tag{3}$$

In Eq (3), we summarise all characteristics in matrices $\mathbf{X}^M$ and $\mathbf{X}^N$ (including years of education and dummies in the error components). Throughout the paper, we assume that immigrants and natives enter the labour market at age $AGE = 20$, and therefore $YSM = 0$, which generates the initial employment probability differential. Then, we increase $AGE$ and $YSM$ by five-year intervals until $AGE = 65$ and $YSM = 45$. We calculate the $REP$ through the life-cycle for alternative levels of each non-cognitive skill of immigrants $NC_p^M$ and natives $NC_p^N$. In our analysis, we keep the non-cognitive skill levels of natives $NC_p^N$ either at the average level $\overline{NC_p^N}$ or the same level as the immigrants $NC_p^N = NC_p^M$. As we obtain a very large set of results in each estimation, we present results with a series of graphics drawn over the life-cycle of immigrants and natives. To summarise the overall effect of each non-cognitive skill on the employment probability integration, we present the *mean lifetime employment probability differential* (*MLD*) of immigrants. It is calculated by averaging the employment probability differential for each year since migration at five-year intervals (10 evaluation points) as

$$MLD(NC_p^M, NC_p^N) = \frac{1}{10} \sum_{a=0}^{9} (E[d^M|AGE = 20 + 5a, YSM = 5a, NC_p^M|\overline{\mathbf{X}}^M]$$
$$-E[d^N|AGE = 20 + 5a, NC_p^N|\overline{\mathbf{X}}^N]). \tag{4}$$

We interpret this measure as an indicator of the relative success of immigrants through their life-cycle in the host county. It is the average lifetime employment probability differential of an immigrant and natives with a particular type and level of non-cognitive skill. *MLD* is expected to be negative, and a lower negative value indicates a better employment probability integration outcome. To measure the sampling variability, we use the delta method to calculate the standard error of $MLD(NC_p^M, NC_p^N)$ and use them while drawing the 90% confidence intervals.

## Results

We first investigate the benchmark level of integration and calculate the overall *MLD* of an average immigrant compared to an average native. Second, we compare the employment probabilities of immigrants with high and low levels of each non-cognitive skill to those of i) an average native in all characteristics, including non-cognitive skills $\overline{NC_p^N}$ and ii) a native with the same non-cognitive skill type and level $NC_p^N = NC_p^M$ while holding all other characteristics at the mean levels. Third, we investigate the observed heterogeneity and robustness of our results. Finally, we focus on whether years of education and non-cognitive skills are substitutes or complements of each other.

### Immigrants' integration in Germany

**Benchmark results.** As the first step, we conduct an analysis to determine the average employment integration levels of immigrants in Germany to generate a comparison benchmark for the heterogeneity analysis below. The model specifications in Eqs (1) and (2) are estimated with the correlated random-effects linear probability model. The benchmark life-cycle employment probabilities of an average immigrant and native are presented in Fig 2. The figure presents the life-cycle predicted employment probabilities (Panel A) of comparable

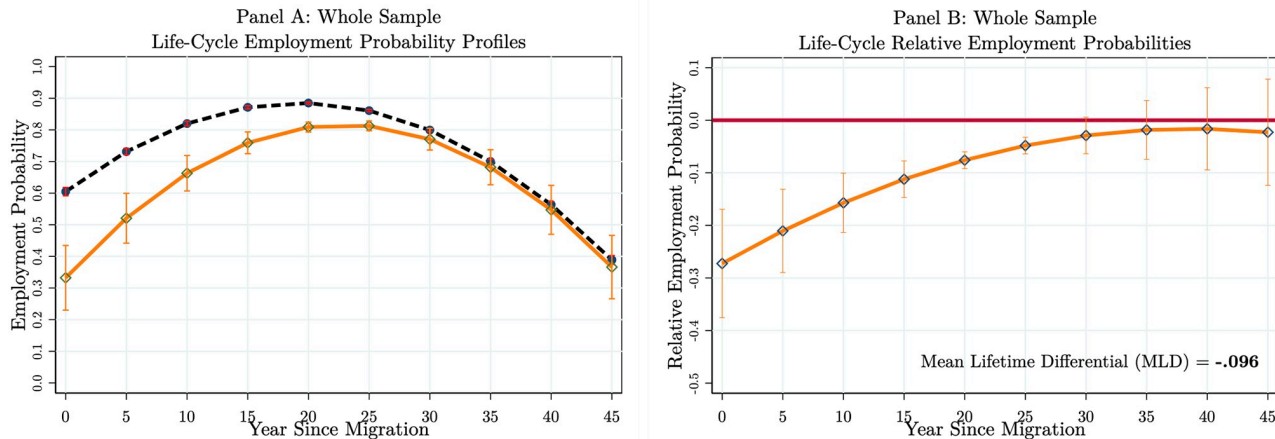

**Fig 2. Benchmark results: Employment probability integration of an average immigrant.** Authors' own calculations from SOEP (1984–2013). The figures are obtained by predicting the employment probabilities using the parameter estimates of the correlated random-effects linear probability model for immigrants and natives. The predicted probabilities of natives are presented with the dashed black curve, while immigrants' predicted employment probabilities are presented with the smooth orange curve (Panel A). The relative employment probabilities (Panel B) are calculated using Eq (3). The mean lifetime employment probability differential (i.e., *MLD*) is presented in the right-bottom corner of Panel B. The 90% confidence intervals are represented as vertical lines. We use the delta method to calculate the standard errors. The model specification allows for the full set of observed characteristics for both immigrants and natives (see Table 1).

immigrants and natives by increasing the years of stay in the host country from 0 (i.e., both immigrants and natives are aged 20) to 45 (i.e., both immigrants and natives are aged 65) with five-year intervals of the year since migration. Panel (B) gives the *relative* employment probabilities of immigrants, i.e. employment probability differentials—measured in percentage points when multiplied by 100. Holding all other variables constant at their mean levels, the relative employment probability of an average immigrant upon entry to the labour market is $REP(AGE = 20, YSM = 0, \overline{NC}_p^M, \overline{NC}_p^N) = -0.274$. It means that an average immigrant enters the labour market with 27.4 percentage points employment probability disadvantage. We then calculate the life-cycle relative employment probabilities (Panel B) at different ages and year since migration combinations using Eq (3).

Panels of Fig 2 suggest that there is a textbook-fashion integration process for an average immigrant in Germany. To calculate the 'upper bound' of the total years to complete integration, we present the 90% confidence intervals (vertical lines) drawn using the standard errors calculated for each evaluation point. Immigrants improve their host country-specific human capital and reduce the initial employment probability differential over time. The 90% confidence intervals of the relative employment probabilities (Panel B, Fig 2) cross the 0 line (red line) about 25–30 years after arrival. We interpret this result as follows: an average immigrant who enters the labour market with about 27.4 percentage points employment probability disadvantage catches up with the employment probability of a comparable average native about 25–30 years after arrival (this is 5–10 years shorter when we use 99% confidence intervals). After that, the life-cycle employment probability profiles (Panel A) become parallel and statistically identical. The integration measure, *MLD*, presented at the right-bottom corner of Panel B, suggests that the mean lifetime employment probability differential is about −9.6 percentage points between comparable immigrants and natives. Calculating the relative employment probabilities for each year after arrival is a highly time-consuming task. Nevertheless, to check the sensitivity of the benchmark degree of integration, we have also calculated the measures for 46 evaluation points each year after arrival. The result hardly differs. Another initial check

is also conducted with respect to estimators. The pooled probit model estimates generate highly similar results (see also Fig A3 in S1 File) with about *MLD* = −9.3 percentage points.

## Non-cognitive skills and employment integration

**Comparing immigrants with an average native.**   We begin to investigate the returns of non-cognitive skills on the employment probabilities of immigrants in comparison to those of natives. That is, we investigate the heterogeneity of the benchmark mean lifetime employment probability differential (*MLD* = 9.6 percentage points) by the non-cognitive skills of immigrants and natives. In our first scenario, we compare an immigrant with a low and high level of each non-cognitive skill type with a native having an *average* level of these skills ($REP(AGE, YSM, NC_p^M, \overline{NC_p^N})$). To this end, we first estimate the specifications in Eqs (1) and (2) with the correlated random-effects linear probability model and then predict the life-cycle employment probabilities of immigrants with the low and high values of each skill type, while holding all other characteristics for both immigrants and natives at their mean levels. To define the low and high levels, we use the $1^{th}$ and $4^{th}$ quartile values of each non-cognitive skill type.

The life-cycle relative employment probability profiles obtained from the first scenario are given in Fig 3 for each type of non-cognitive skill (the life-cycle employment probabilities of immigrants and natives are presented in Fig A4 in S1 File). The *MLD* among immigrants with the low ($1^{th}$ quartile) and high levels ($4^{th}$ quartile) of each non-cognitive skill are presented at the right bottom corner of each panel. First of all, there are alternative integration patterns generated by each non-cognitive skill type. Second, there is substantial heterogeneity in the *MLD* generated by the low and high levels of each skill type. For instance, immigrants with a higher (dark blue curve with triangles) and a lower (orange curve with diamonds) level of extroversion (the first graph in Fig 3 and Fig A4 in S1 File) enter into the host country labour market with almost the same employment probability disadvantage. Yet, immigrants scoring high on extroversion (highly communicative and social immigrants) experience a faster increase in their marginal rates of integration, and these immigrants are able to *fully* catch up with the employment probability parity of an average native in 20–25 years after arrival (shorter than 5–10 years compared to benchmark results, Fig 2). Immigrants who score low on extroversion (orange curves) obtain large employment probability differentials throughout their immigration experience. These immigrants' integration is weak and very slow, and they are not able to attain full integration in their lifetime. The *MLD* also suggests that immigrants who score high on extroversion achieve a better integration (−5.9 percentage points) compared to immigrants who score low on extroversion (−11.9 percentage points).

We observe a similar pattern for all other non-cognitive skill types except for agreeableness. A higher level of each skill type relates to a higher speed of integration during the first years after arrival, positive marginal integration rates for longer periods, and finally, a better overall integration outcome over the life-cycle. The largest return of non-cognitive skills is obtained by immigrants who score high on emotional stability (*MLD* = −3.7 vs. −15.0 percentage points) and conscientiousness (*MLD* = −5.3 vs. −16.5 percentage points). Finally, we find that the immigrants who score high on agreeableness experience a relatively higher mean lifetime differential (*MLD* = −13.0 percentage points) compared to immigrants who score low on agreeableness (*MLD* = −8.5 percentage points). This result is consistent with our predictions that agreeable immigrants (evading conflicts and seeking cooperation) might not be successful if they face a high labour market competition with natives. One final remark is that the effect of non-cognitive skills appears to be stronger after 15–20 years of arrival. This result is also consistent with our predictions that immigrants might be able to productively use their non-cognitive skills only after developing a particular level of host country-specific skills. Overall,

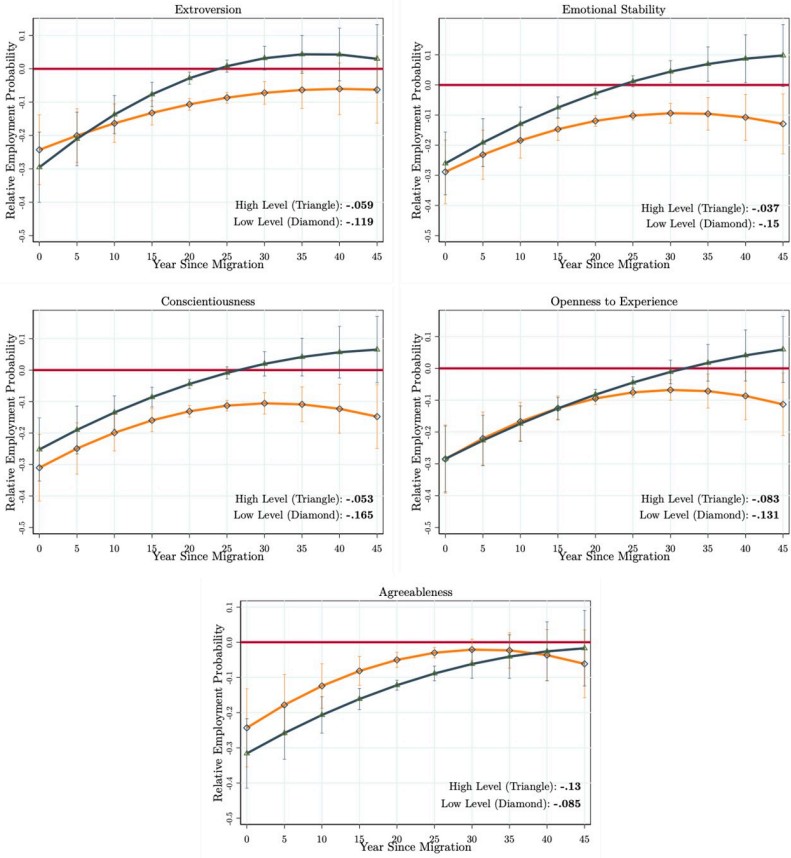

**Fig 3. Comparing immigrants with an average native.** Authors' own calculations from the SOEP (1984–2013). The model specifications estimated are given in Eqs (1) and (2). They are estimated with the correlated random-effects linear probability model. The figures give only the relative employment probabilities calculated for five-year intervals of after arrival. The red horizontal line passes from zero. The dark blue curve (with triangles) presents the relative employment probability profile for immigrants scoring high on each personality type, while the orange curves (with diamonds) give the same for the low levels of each personality type. The *MLD* for high and low levels of each personality type are given in the right bottom corner of each figure. We use the delta method to calculate the standard errors. The 90% confidence intervals are represented as vertical lines.

the non-cognitive skills of immigrants are productive in their integration process when we compare their performance with that of an average native.

**Comparing immigrants and natives with the same skill type and levels.** In our second scenario, we investigate the immigrants' integration compared to natives with the same level of each non-cognitive skill type ($REP(AGE, YSM, NC_p^M, NC_p^N = NC_p^M)$). This comparison assumes that immigrants and natives sort in the labour market by their non-cognitive skill types and levels to compete for jobs (e.g., a highly conscientious immigrant versus a highly conscientious native). Note also that if the returns of non-cognitive skills for the immigrants and natives are the same, then we obtain no differences in the level of integration for the low and high levels of each skill type. As in the previous case, we calculate the life-cycle relative employment probabilities by comparing immigrants and natives with the low and high levels of each skill type for 45 years after immigration.

The results are presented in the panels of Fig 4 (the life-cycle employment probabilities of immigrants and natives are presented in Fig A5 in S1 File). Even though the differences are

smaller, we find that there are differential returns of non-cognitive skills on immigrants' employment integration when compared to natives with the same level of each skill type. Similar to the previous scenario, immigrants who score high on extroversion are also able to fully integrate into the employment level of an average native who also scores high on extroversion. This process takes about 15–20 years. Yet, immigrants who score low on extroversion experience very high employment probability differentials, and they are not able to fully integrate. The mean lifetime differential of immigrants with a high level of extroversion is about $MLD = -5.9$ while it is about $MLD = -10.9$ percentage points for the immigrants who score low on extroversion. Even though we observe similar patterns to the ones reported in Fig 3, comparing immigrants and natives with the same level of non-cognitive skills generates a lower level of relative returns. While we find no relative return of emotional stability and agreeableness, immigrants who score high on extraversion, conscientiousness, and openness to experience obtain 3–5 percentage points lower mean lifetime employment probability differential.

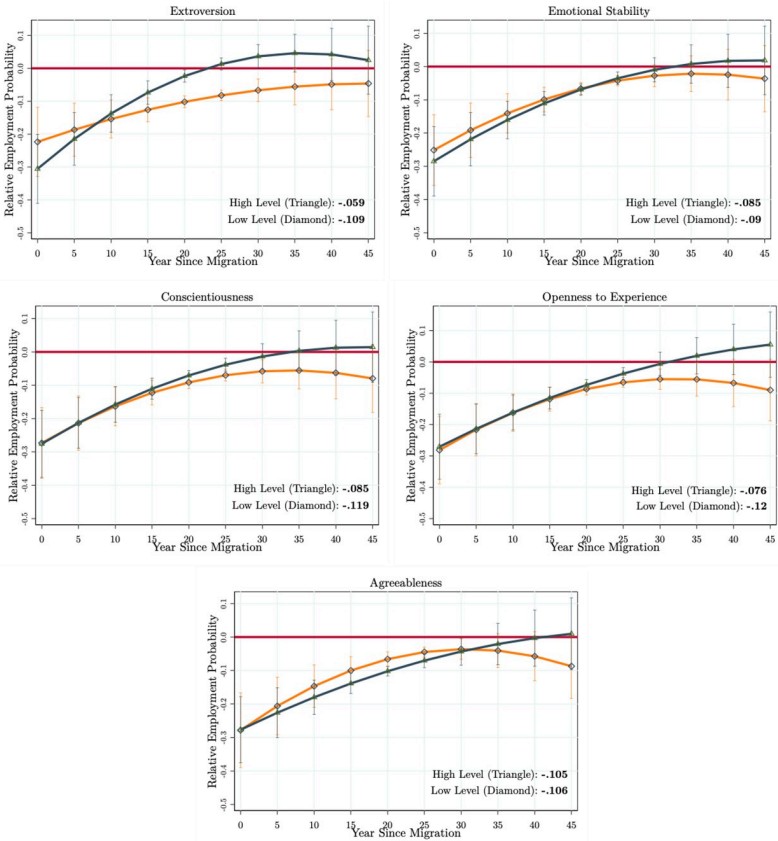

**Fig 4. Comparing immigrants and natives with the same non-cognitive skill types and levels.** Authors' own calculations from the SOEP (1984–2013). The model specifications estimated are given in Eqs (1) and (2). The specifications are estimated with the correlated random-effects linear probability model. The figures present the relative employment probabilities calculated for every five years since arrival (see Fig A5 in S1 File for the life-cycle employment probability profiles of immigrants and natives). The red horizontal line passes from zero. The dark blue curve (with triangles) gives the relative employment probability profile of immigrants scoring high levels on each non-cognitive skill type, while the orange curves (with diamonds) give the same for the low levels. The MLD for high and low levels are given in the right bottom corner of each figure. We use the delta method to calculate the standard errors. The 90% confidence intervals are represented as vertical lines.

## Observed heterogeneity

A set of heterogeneity analyses is conducted and presented in Table 1. To save space and summarise a large collection of results, we only present the *MLD* and the standard errors to assess the statistical significance. First two rows reproduce the *MLD* for the case where we compare the immigrants with low and high non-cognitive skills with that of an average native (Row I, A) and a native with the same type and level of non-cognitive skills (Row I, B). To calculate the observed heterogeneity in *MLD*, we estimate the integration outcomes for several groups of immigrants. There are two sets of results for each heterogeneity analysis split as (A) and (B). The results in (A) present the effect of immigrants' non-cognitive skills compared to an average native, while the results in (B) give the effects by comparing the immigrants and natives with the same type and level of each non-cognitive skill as in Figs 3 and 4.

**Is there a gender difference?.** The results split for male and female immigrants are given in Rows II and III of Table 1. One important result is that the benchmark *MLD* of male immigrants is small in magnitude (−5 percentage points) and statistically insignificant in the whole sample. The key non-cognitive skill for the degree of integration for male immigrants is conscientiousness, followed by emotional stability. Row II(A) suggests that male immigrants who score low on conscientiousness experience a statistically significant *MLD* which is about −12.8 percentage points, while the *MLD* of male immigrants who score high on conscientiousness is as low as −1.1 percentage points. Males with lower emotional stability also have a relatively large *MLD* = −7.8 percentage points, but the differential is reduced to *MLD* = −2.6 percentage points among males with higher emotional stability. Comparing male immigrants with natives having the same non-cognitive skills (Row II(B)) suggests that the differential returns of non-cognitive skills for male immigrants are relatively small. The result implies that the labour market equally values the non-cognitive skills of male immigrants and natives.

The results for female immigrants given in Rows III(A) and (B) of Table 1 tell a completely different story. First of all, the benchmark results (Figs 2–4) for the degree of integration are driven by employment outcomes of female immigrants. The benchmark *MLD* for the whole female immigrant group is −12.6 percentage points, which is statistically significant at the conventional levels. Second, compared to an average female native, all non-cognitive skill types have a large effect on the overall degree of integration (Row III(A), Table 1). Third, except for agreeableness, female immigrants who score high on each non-cognitive skill type experience a lower *MLD*. For instance, female immigrants with a low conscientiousness experience −21.0 percentage points mean lifetime differential compared to an average female native, while female immigrants with a high level of conscientiousness experience only a −7.0 percentage points differential. Finally, Row III(B) compares female immigrants with the female natives having the same level of each non-cognitive skill. There are still sizeable heterogeneity in *MLD* for extroversion (low −15.1 vs. high −7.5 percentage points), conscientiousness (low −16.2 vs. high −11.4 percentage points), and openness to experience (low −18.1 vs. high −10.0 percentage points).

**The country of origin.** Another heterogeneity analysis is conducted among immigrants from the EU (Row IV, Table 1) and the non-EU countries (Row V, Table 1). First of all, we find that the *MLD* of the EU immigrants is lower (−5.9 percentage points) than that of non-EU immigrants (−14.2 percentage points), and it is statistically significant only among the latter group. Compared to an average native (Row IV(A) and V(A)), non-cognitive skills are productive on the levels of integration among both immigrant groups. The EU immigrants who score low on emotional stability and conscientiousness are penalised, and they experience a significantly higher *MLD* (Row IV(A)). Yet, all non-cognitive skills are essential for the integration level of non-EU immigrants compared to an average native (Row V(A)). Scoring low

**Table 1. Observed heterogeneity.**

| Benchmark Results | | Immigrants' #Obs. Natives' #Obs | Whole Sample | Big-5 Personality Charactetistics | | | | | | | | | |
|---|---|---|---|---|---|---|---|---|---|---|---|---|---|
| | | | | Extroversion | | Emotional Stability | | Consicientoeness | | Openness to Experience | | Aggreableness | |
| | | | | | | Low | High | | | Low | High | Low | High |
| I Compared to average native | (A) | 28,582 / 208,064 | -0.096 *** / 0.035 | -0.119 *** / 0.035 | -0.059 * / 0.036 | -0.150 *** / 0.035 | -0.037 / 0.036 | -0.165 *** / 0.036 | -0.053 / 0.035 | -0.131 *** / 0.035 | -0.083 ** / 0.036 | -0.085 ** / 0.036 | -0.130 *** / 0.036 |
| Compared to same level personality | (B) | | | -0.109 *** / 0.035 | -0.059 * / 0.036 | -0.090 ** / 0.035 | -0.085 ** / 0.036 | -0.119 *** / 0.036 | -0.085 ** / 0.036 | -0.120 *** / 0.036 | -0.076 ** / 0.036 | -0.106 *** / 0.036 | -0.105 *** / 0.036 |
| II II Male | (A) | 13,218 / 108,192 | -0.050 / 0.048 | -0.059 / 0.049 | -0.055 / 0.049 | -0.078 / 0.049 | -0.026 / 0.050 | -0.128 *** / 0.049 | -0.011 / 0.049 | -0.072 / 0.049 | -0.041 / 0.050 | -0.075 / 0.049 | -0.042 / 0.049 |
| | (B) | | | -0.047 / 0.049 | -0.055 / 0.050 | -0.048 / 0.049 | -0.048 / 0.050 | -0.078 / 0.049 | -0.044 / 0.049 | -0.058 / 0.049 | -0.046 / 0.050 | -0.079 / 0.049 | -0.039 / 0.049 |
| III III Female | (A) | 15,369 / 99,872 | -0.126 ** / 0.050 | -0.179 *** / 0.051 | -0.053 / 0.052 | -0.154 *** / 0.050 | -0.075 / 0.051 | -0.210 *** / 0.051 | -0.070 / 0.051 | -0.196 *** / 0.051 | -0.091 / 0.051 | -0.115 ** / 0.051 | -0.147 *** / 0.051 |
| | (B) | | | -0.151 *** / 0.051 | -0.075 / 0.052 | -0.105 ** / 0.051 | -0.119 ** / 0.051 | -0.162 *** / 0.051 | -0.114 ** / 0.051 | -0.181 *** / 0.051 | -0.100 * / 0.051 | -0.126 ** / 0.051 | -0.136 *** / 0.051 |
| IV IV EU immigrants | (A) | 14,518 / 208,064 | -0.059 / 0.050 | -0.058 / 0.050 | -0.047 / 0.050 | -0.123 ** / 0.050 | 0.001 / 0.050 | -0.109 ** / 0.050 | -0.022 / 0.050 | -0.068 / 0.050 | -0.082 / 0.050 | -0.054 / 0.051 | -0.083 * / 0.050 |
| | (B) | | | -0.048 / 0.050 | -0.048 / 0.050 | -0.063 / 0.050 | -0.046 / 0.050 | -0.063 / 0.051 | -0.054 / 0.050 | -0.057 / 0.050 | -0.076 / 0.050 | -0.075 / 0.051 | -0.058 / 0.050 |
| V V Non-EU immigrants | (A) | 14,064 / 208,064 | -0.142 *** / 0.052 | -0.190 *** / 0.052 | -0.074 / 0.054 | -0.192 *** / 0.052 | -0.079 / 0.053 | -0.219 *** / 0.052 | -0.100 * / 0.053 | -0.203 *** / 0.052 | -0.093 * / 0.053 | -0.113 ** / 0.053 | -0.190 *** / 0.053 |
| | (B) | | | -0.180 *** / 0.052 | -0.075 / 0.054 | -0.132 ** / 0.052 | -0.126 ** / 0.053 | -0.173 *** / 0.053 | -0.132 ** / 0.053 | -0.193 *** / 0.052 | -0.086 / 0.053 | -0.135 ** / 0.053 | -0.165 *** / 0.053 |
| VI VI Guest Workers | (A) | 14,673 / 208,064 | -0.170 *** / 0.049 | -0.206 *** / 0.049 | -0.139 *** / 0.051 | -0.237 *** / 0.050 | -0.120 ** / 0.051 | -0.223 *** / 0.050 | -0.132 *** / 0.049 | -0.191 *** / 0.049 | -0.179 *** / 0.051 | -0.139 *** / 0.050 | -0.232 *** / 0.049 |
| | (B) | | | -0.197 *** / 0.049 | -0.140 *** / 0.051 | -0.178 *** / 0.050 | -0.137 *** / 0.051 | -0.177 *** / 0.050 | -0.123 ** / 0.049 | -0.181 *** / 0.049 | -0.112 ** / 0.051 | -0.160 *** / 0.050 | -0.207 *** / 0.049 |

Authors' own calculations from SOEP waves from 1984 to 2013. The table reports results from our robustness analysis. The measure reported is the mean lifetime employment probability differential, *MLD*, of immigrants by the low and high values of each non-cognitive skills among alternative groups of immigrants, personality measures observed in different waves, and local labour market characteristics. The models are estimated based on the correlated random-effects linear probability model specification. The specifications control for the full set of control variables. The standard errors presented in the parenthesis are obtained with the delta method.

*, **, and *** indicate significance levels at 10%, 5%, and 1%, respectively.

on any type of non-cognitive skill leads to about 5–15 percentage points higher *MLD* among these immigrants. Compared to natives with the same level of non-cognitive skills, there is no differential return among the EU immigrants (Row IV(B)), while extroversion and openness to experience are very important for the level of employment probability integration for the non-EU immigrants (Row V(B)).

To further investigate the heterogeneity with respect to the country of origin, we also focus on immigrants from the *guest worker* countries (Row VI, Table 1). These immigrants arrived in Germany during the 1970s and constituted a significant share of the immigrant stock. These countries are mainly Turkey, Greece, Italy, Spain, Portugal, Morocco and former Yugoslavian countries (see Table A3 in S1 File for the descriptive statistics). We obtain the largest differentials among this group of immigrants. Their overall mean lifetime differential is about *MLD* = −17.4 percentage points. Except for openness to experience, immigrants who score low on any type of non-cognitive skill obtain about *MLD* = −20 percentage points differential (Row VI (A)), which is up to two times larger compared to guest workers who score high on these skills. This result is reversed for agreeableness. Compared to natives with the same type and level of non-cognitive skills (Row VI(A)), the relative returns of non-cognitive skills among these immigrants are also sizeable for all types of non-cognitive skills, with about 5–10 percentage points lower mean lifetime differentials.

## Robustness

**Self-selection.** Do immigrants self-select into the labour market of the host country with their non-cognitive skills? Previous research suggests that intrinsic characteristics, including risk-taking, openness to experience, and extroversion, might relate to immigration decisions [63]. In our case, because we observe immigrants ex-post, we are not able to capture the effect of self-selection on our results. Nevertheless, we check the robustness with respect to self-selection by the non-cognitive skills by using the "arrival age" of immigrants. The conjecture is that the likelihood of self-selection by non-cognitive skills is lower among those who arrive younger. This idea is supported by personality psychology suggesting that personality characteristics are formed during adolescence (age 10–19) and stay relatively stable later (e.g., [12, 48]). Thus, we conduct our analysis among immigrants who arrived in the host country at 19 years old and younger versus those older than 19 to investigate the potential self-selection. The results are presented in Rows II and III of Table 2. As in the previous table, we present the comparisons with an average native (A) and with the native having the same level of each non-cognitive skill type (B). Rows II(A) and III(A) suggest that there are no sizeable differences between immigrants who arrived younger or older. A similar result is also obtained when we make the comparison between immigrants and natives with the same non-cognitive skill type and level (Rows II(B) and III(B), Table 2). Among unreported results, we estimate models using immigrants who arrived at 13 or younger and those older than 13. The sample size among immigrants who arrived at 13 and younger is relatively small (6,431 immigrants-year observations). Nevertheless, we find a similar effect of non-cognitive skills on the integration outcomes of immigrants.

**Non-random returns.** When immigrants with a particular non-cognitive skill type return to their home country, the rest of the sample might lead to selectivity bias. Although the actual number of returns from Germany is small and panel attrition is also found to be weakly related to the non-cognitive skills (e.g., [52]), it is worth checking whether there is a substantial effect of such sample selectivity. Our strategy is as follows. We first identify the individuals exiting the sample with their reason for moving abroad. We then delete the existing information about these immigrants in the earlier waves of the panel based on their return status. Simple

**Table 2. Robustness.**

| Benchmark Results | | Immigrants' #Obs. | Natives' #Obs | Whole Sample | Big-5 Personality Charactetistics | | | | | | | | | |
|---|---|---|---|---|---|---|---|---|---|---|---|---|---|---|
| | | | | | Extroversion | | Emotional Stability | | Conscientoeness | | Openness to Experience | | Agreableness | |
| | | | | | Low | High | Low | High | Low | High | Low | High | Low | High |
| I Compared to average native Compared to same level of personality | (A) | 28,582 | 208,064 | -0.096 (0.035) *** | -0.119 (0.035) *** | -0.059 (0.036) * | -0.150 (0.035) *** | -0.037 (0.036) | -0.165 (0.036) *** | -0.053 (0.035) | -0.131 (0.035) *** | -0.083 (0.036) ** | -0.085 (0.036) ** | -0.130 (0.036) *** |
| | (B) | | | | -0.109 (0.035) *** | -0.059 (0.036) * | -0.090 (0.035) ** | -0.085 (0.036) ** | -0.119 (0.036) *** | -0.085 (0.036) ** | -0.120 (0.036) *** | -0.076 (0.036) ** | -0.106 (0.036) *** | -0.105 (0.036) *** |
| II Arrival age (older than 20) | (A) | 14,431 | 208,064 | -0.111 (0.050) ** | -0.133 (0.051) *** | -0.089 (0.051) * | -0.183 (0.050) *** | -0.039 (0.052) | -0.192 (0.051) *** | -0.077 (0.051) | -0.155 (0.051) *** | -0.077 (0.052) | -0.104 (0.051) ** | -0.140 (0.052) *** |
| | (B) | | | | -0.123 (0.051) ** | -0.089 (0.051) * | -0.123 (0.051) ** | -0.086 (0.052) * | -0.146 (0.051) *** | -0.108 (0.051) ** | -0.144 (0.051) *** | -0.071 (0.052) | -0.126 (0.051) ** | -0.115 (0.052) ** |
| III Arrival age (20 and younger) | (A) | 14,151 | 208,064 | -0.106 (0.060) * | -0.134 (0.060) ** | -0.051 (0.063) | -0.147 (0.060) ** | -0.048 (0.063) | -0.159 (0.062) ** | -0.064 (0.061) | -0.139 (0.060) ** | -0.102 (0.062) | -0.082 (0.063) | -0.155 (0.060) *** |
| | (B) | | | | -0.124 (0.060) ** | -0.051 (0.063) | -0.087 (0.061) | -0.095 (0.063) | -0.113 (0.062) * | -0.096 (0.061) | -0.129 (0.060) ** | -0.095 (0.062) | -0.103 (0.063) * | -0.131 (0.060) ** |
| IV Without actual returners | (A) | 25,245 | 208,064 | -0.087 (0.037) ** | -0.108 (0.037) *** | -0.051 (0.038) | -0.136 (0.037) *** | -0.026 (0.037) | -0.144 (0.037) *** | -0.049 (0.037) | -0.112 (0.037) *** | -0.081 (0.037) ** | -0.075 (0.037) ** | -0.123 (0.037) *** |
| | (B) | | | | -0.098 (0.037) *** | -0.051 (0.038) | -0.076 (0.037) ** | -0.073 (0.037) * | -0.099 (0.037) *** | -0.081 (0.037) ** | -0.121 (0.037) *** | -0.074 (0.037) ** | -0.096 (0.038) ** | -0.098 (0.037) *** |
| V Stability of personality (only 2005) | (A) | 28,582 | 208,064 | -0.096 (0.035) *** | -0.122 (0.035) *** | -0.063 (0.036) * | -0.141 (0.035) *** | -0.043 (0.036) | -0.169 (0.036) *** | -0.052 (0.035) | -0.137 (0.035) *** | -0.096 (0.036) *** | -0.088 (0.036) ** | -0.121 (0.035) *** |
| | (B) | | | | -0.110 (0.035) *** | -0.067 (0.036) * | -0.097 (0.035) *** | -0.088 (0.036) ** | -0.124 (0.036) *** | -0.083 (0.035) ** | -0.129 (0.035) *** | -0.080 (0.036) ** | -0.103 (0.036) *** | -0.103 (0.036) *** |
| VI Stability of personality (only 2013) | (A) | 28,582 | 208,064 | -0.093 (0.035) *** | -0.114 (0.035) *** | -0.065 (0.036) * | -0.139 (0.035) *** | -0.033 (0.036) | -0.143 (0.036) *** | -0.042 (0.035) | -0.128 (0.035) *** | -0.090 (0.036) ** | -0.082 (0.036) ** | -0.133 (0.036) *** |
| | (B) | | | | -0.108 (0.035) *** | -0.068 (0.036) * | -0.090 (0.035) ** | -0.085 (0.036) ** | -0.113 (0.036) *** | -0.069 (0.036) * | -0.122 (0.035) *** | -0.088 (0.036) ** | -0.099 (0.036) *** | -0.109 (0.036) *** |
| VII Local Market Characteristics (ROR) | (A) | 22,462 | 170,074 | -0.099 (0.040) ** | -0.120 (0.040) *** | -0.059 (0.041) | -0.152 (0.040) *** | -0.038 (0.041) | -0.169 (0.041) *** | -0.060 (0.040) | -0.128 (0.041) *** | -0.091 (0.041) ** | -0.087 (0.041) ** | -0.130 (0.041) *** |
| | (B) | | | | -0.111 (0.041) *** | -0.060 (0.041) | -0.093 (0.040) ** | -0.084 (0.041) ** | -0.120 (0.041) *** | -0.094 (0.041) ** | -0.118 (0.041) *** | -0.074 (0.041) * | -0.107 (0.041) *** | -0.107 (0.041) *** |

Authors' own calculations from SOEP waves from 1984 to 2013. The table reports results from our robustness analysis. The measure reported is the mean lifetime employment probability differential, *MLD*, of immigrants by the low and high values of each non-cognitive skills among alternative groups of immigrants, personality measures observed in different waves, and local labour market characteristics. The models are estimated based on the correlated random-effects linear probability model specification. The specifications control for the full set of control variables. The standard errors presented in the parenthesis are obtained with the delta method.

\*, \*\*, and \*\*\* indicate significance levels at 10%, 5%, and 1%, respectively.

statistics suggest that 11.6% of the immigrants in the sample moved abroad, constituting 3,321 immigrant-year observations to be deleted from the sample. Excluding returned individuals from the estimation sample generates $MLD = -8.7$ percentage points overall differential (Row IV, Table 2), which is about one percentage point less than that of the baseline (Row I, Table 2). There is no particularly prominent effect of non-random returns on the relative sizes of the $MLD$ found for the low and high levels of each non-cognitive skill type. Thus, the main results hardly differ for any type of non-cognitive skill.

**Stability of personality.** The literature argues that Big-5 personality characteristics are relatively stable over time and are not strongly affected by life circumstances (e.g., [48, 51, 64]). Throughout this study, we also assume that these characteristics are stable among immigrants and natives. As we have reported in Figs A1 and A2 in S1 File, there are also no significant changes in the mean levels and in the distributions of each skill type among immigrants who stayed shorter (less than 20 years) and longer (20 and more) in the host country. Nevertheless, to investigate this point further, we use Big-5 personality information obtained during different points in the life of immigrants and natives. The dataset includes information on the non-cognitive skills in waves 2005, 2009, and 2013. We now estimate our model specification using waves 2005 and 2013 as two distinct non-cognitive skill information obtained eight years apart for the same individual. At least for a period, we can show whether the results are stable with respect to non-cognitive skill measures obtained for different years since migration/age of immigrants and natives. Rows V and VI of Table 2 suggest that the results are highly consistent with those of the main results presented in Figs 3 and 4.

**Estimators and model specifications.** To check the robustness with respect to estimators, we estimate a pooled probit model. The model specifications control for all variables used in Eqs (1) and (2), including within means of time-variant variables. The relative employment probabilities in comparison to average natives obtained from the pooled probit model specification are given in Fig A6 in S1 File. Even though the estimated profiles look somewhat different, the patterns support the previous results that immigrants with a higher level of each non-cognitive skill type are more successful in the labour market, while immigrants with a lower level of each skill type are not able to integrate fully (except for agreeableness; the result is reversed in that case). Similar to the baseline, $MLD$ of immigrants with a high level of non-cognitive skill is 5–15 percentage points lower. Among the unreported results, we also compare immigrants and natives with the same level of each non-cognitive skill type as in Fig 4. There are differential returns to extroversion and openness to experience in the integration process, generating a 3–5 percentage point lower lifetime employment probability differential.

Finally, we conduct a specification check by allowing heterogeneity in the local labour markets with respect to local overall unemployment rates among people aged 15–65, unemployment rates among immigrants, and the local GDP per capita. The idea is that the local labour markets might differentially affect the immigrants and natives at the year of observation. Thus, these characteristics could be another set of unobserved confounding factors correlating with the integration of immigrants. The data are obtained for 96 ROR regions between 1998 and 2013. The resulting sample size is 22,462 for immigrants and 170,074 for natives. The results presented in Row VI of Table 2 suggest that controlling for local market characteristics in the model specifications does not significantly relate to the main findings.

## Standard human capital measures vs. non-cognitive skills: Are they substitutes or complements?

Finally, we focus on how non-cognitive skills relate to the employment probability integration among immigrants with a lower and higher level of the formal years of education. To this end,

we split our sample by the *International Standard Classification of Education* (ISCED hereafter) score of immigrants and natives. There are six skill classifications in ISCED given as follows: 1) *no or inadequate schooling/in school*; 2) *general elementary*; 3) *middle vocational*; 4) *vocational plus abitur*; 5) *higher vocational*; 6) *higher education*. We group the first three categories as the *low-skill*, i.e., low level of education, and the rest as the *high-skill*, i.e., high level of education (the percentage of high-cognitive skill is 20.2% among immigrants and 30.4% among natives). Then, we analyse the low and high levels of each non-cognitive skill type by splitting the sample into low- and high-skilled immigrants and natives. As there is a large collection of results, we summarise our findings in Figs 5 and 6. First, in Fig 5, we compare immigrants with low and high non-cognitive skills with an average native by splitting the data for low (left panels) and high-skills (right panels). Second, in Fig 6, we compare immigrants and natives with the same non-cognitive skill type and levels split by low- (left panels) and high-skills (right panels). The dark blue curves (with triangles) represent the life-cycle relative employment probability differentials for the high level of each non-cognitive skill, while orange curves (with diamonds) represent the differentials for the low levels.

The results reported in Fig 5 unveil important integration patterns. First of all, the overall effect of non-cognitive skills on employment integration (see Fig 3) is driven by low-skilled immigrants (the left panels of Fig 5). Low-skilled immigrants who score high on extroversion, emotional stability, and conscientiousness integrate faster and obtain better integration outcomes. They also experience a lower *MLD* (right bottom corner of each figure) than low-skilled immigrants who score low on extroversion. While there is no effect of openness to experience, a lower level of agreeableness among low-skilled immigrants implies a better integration outcome. However, the effects of non-cognitive skills are weaker among high-skilled immigrants and natives (the right panels of Fig 5). Even though there are some differences in *MLD*, confidence intervals of the relative employment probabilities intersect at each point of comparison.

Fig 6 also shows that the results are driven by low-skilled (left panels) immigrants (see Fig 4). Comparing immigrants and natives with the same non-cognitive skill levels suggests that a higher level of non-cognitive skills leads to a lower mean lifetime differential among the lower educated immigrants (up to 3–5 percentage points lower *MLD*). In particular, we find that extroversion and, to some extent, openness to experience are the key non-cognitive skills among low-skilled immigrants for their employment probability integration. For instance, low-skilled immigrants who score high on extroversion can catch up with the employment parity of low-skilled natives who score high on extroversion. Overall, we interpret these results as follows: the non-cognitive skills are *substitutes* for the standard human capital measures of the labour market skills in the integration process, as they mainly operate among immigrants with lower skill levels. There is no effect of non-cognitive skills on the integration levels of high-skilled immigrants.

## Conclusion

Integrating the existing stock of immigrants into the labour market and selecting new immigrants who are matched well to the host country's labour market in their observable and unobservable characteristics is very important for Western countries because poor economic success may lead to severe social cohesion and political problems. To this end, to the best of our knowledge, this paper is among the first to explore whether and how non-cognitive skills help first-generation immigrants in their integration process and how standard human capital measures (e.g., formal education) and non-cognitive skills interact as a determinant of immigrants' labour market success in the host country. The analysis uses high-quality panel data

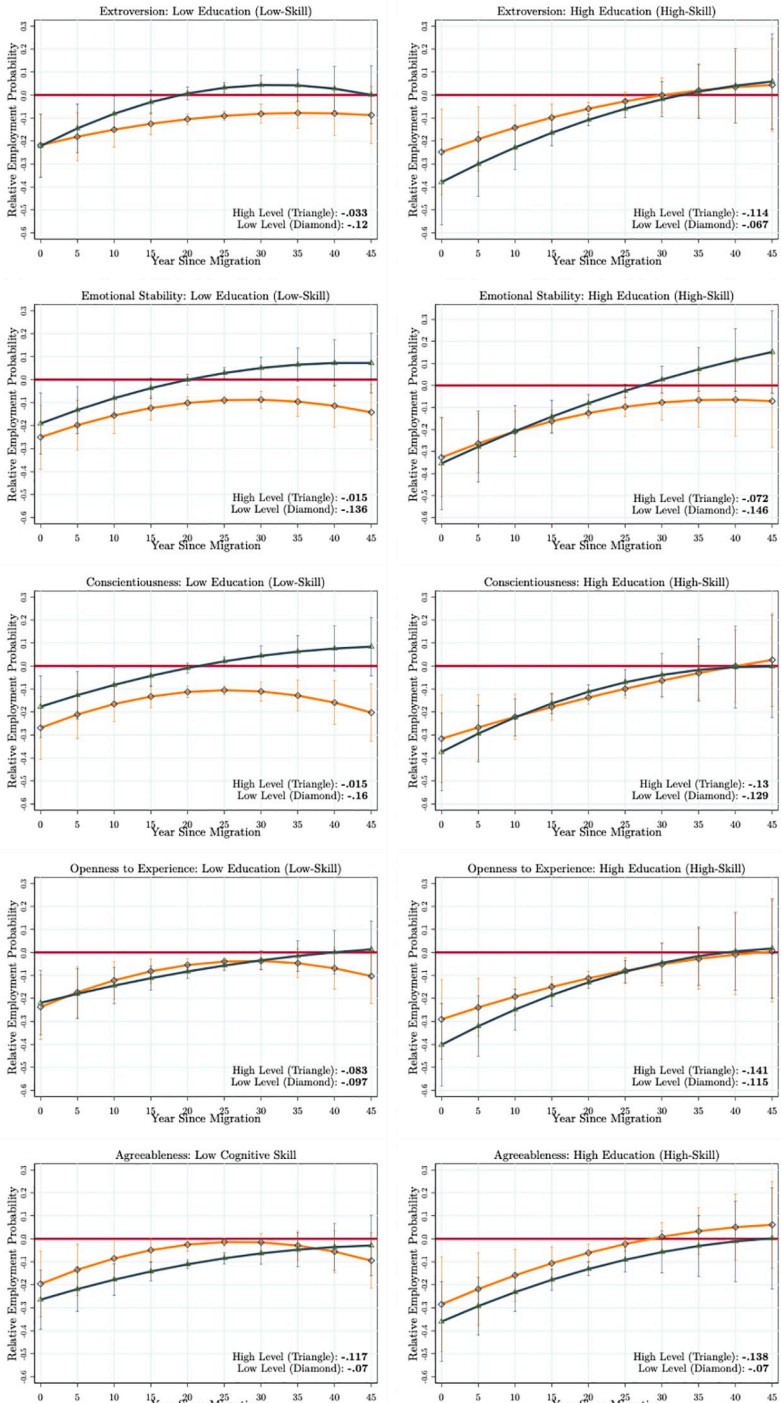

**Fig 5. Years of education vs. non-cognitive skills: Comparing with the average native.** Authors' own calculations from the SOEP (1984–2013). See the note under Figs 3 and 4.

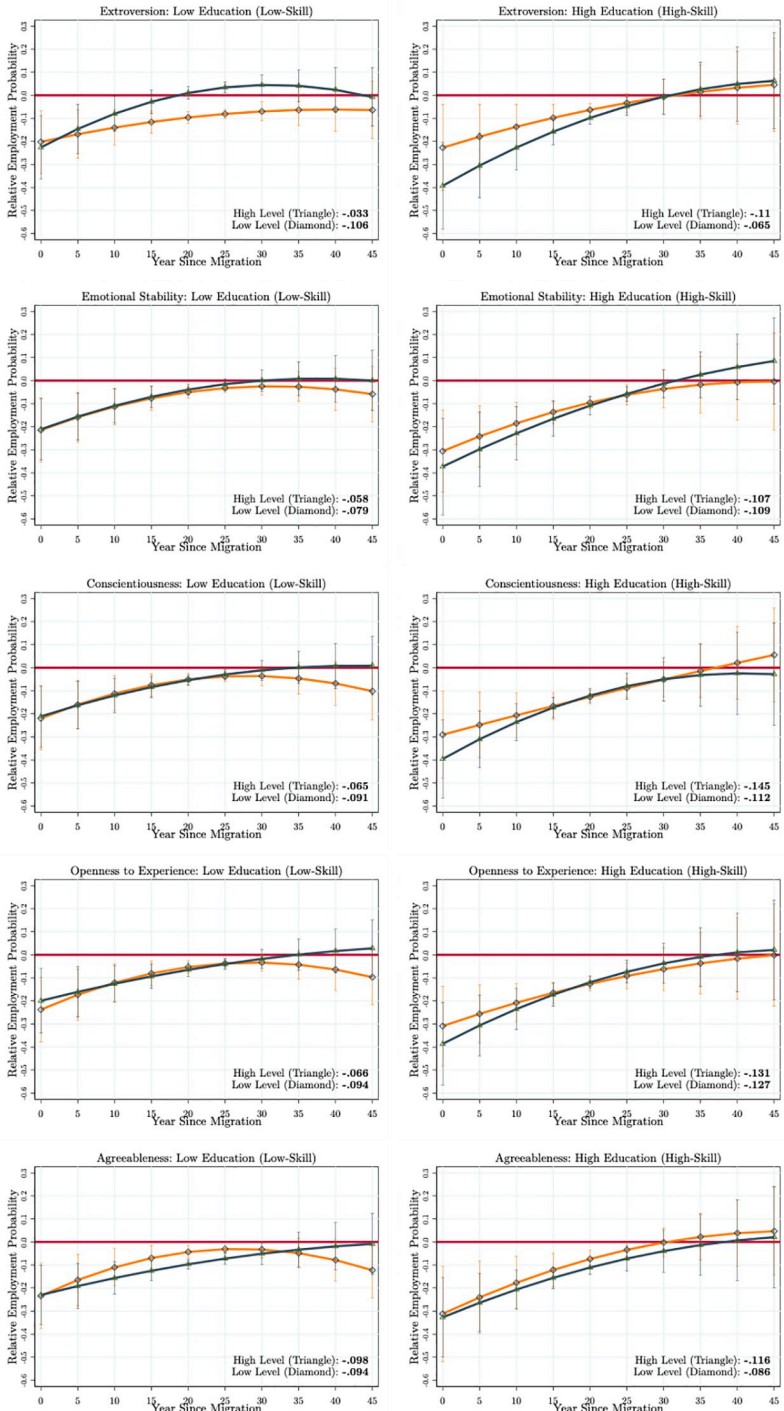

**Fig 6. Years of education vs. non-cognitive skills: Comparing with the same level native.** Authors' own calculations from the SOEP (1984–2013). See the note under Figs 3 and 4.

collected in Germany, one of the developed countries with a large stock of immigrants. Measured with Big-5 personality characteristics, the non-cognitive skills of immigrants are found to be highly important in their employment probability integration process in the host-country labour market.

A detailed summary of the results is as follows. There is a steady but slow integration process in Germany on average. The total number of years to full integration in Germany is about 25–30 years (this result can be considered as the 'upper bound', and it is 5–10 years shorter when we calculate 95–99% confidence intervals). Immigrants experience about a −9.6 percentage points employment probability disadvantage throughout their lifetime, on average. Immigrants who score high on extroversion, emotional stability, conscientiousness, and openness to experience are able to fully reach the employment probability level of an average native. Their mean lifetime employment probability differential is lower by up to 5–15 percentage points. However, the immigrants who score high on agreeableness experience either no effect or a slightly higher mean lifetime differential. Comparing immigrants and natives with the same type and level of non-cognitive skills unveils that the returns to extroversion and, to some extent, openness to experience are higher among immigrants. On average, immigrants who score high on these non-cognitive skills integrate better and experience 3–5 percentage points lower lifetime employment probability differentials. The non-cognitive skills significantly operate only among immigrants with lower education. A lower level of non-cognitive skills (particularly extroversion, i.e., highly sociable, talkative, assertive, and enthusiastic individuals) leads to poor labour market performance, while there is no particular impact of non-cognitive skills on the integration process of highly educated immigrants. We conclude that the non-cognitive skills are 'substitutes' for the standard human capital measures used to capture labour market skills as they are only operating in the integration process of immigrants with a low level of education.

This paper suggests the first evidence on how the non-cognitive skills of immigrants relate to their labour market success in the host country. The results have important methodological and policy implications. First of all, the results suggest that the standard economic integration model, which is mainly based on standard human capital measures, should be enhanced to cover non-cognitive dimensions of immigrants' skills. Second, immigrants might be discouraged from operationalising their non-cognitive skills due to negative sentiments or discrimination in the labour market. Thus, integration policies should focus on specific personality types which are more vulnerable to these labour market constraints. Third, our results suggest that non-cognitive skills have a more prominent effect on the integration outcomes of female immigrants, those from non-EU countries, and immigrants with a low level of education and training. Policies should look beyond formal education and target operationalising the non-cognitive skills of these groups of immigrants for better integration outcomes. One limitation of this paper is that it focuses only on the "employment" integration outcomes of immigrants. Yet, future research should focus on how other dimensions of immigrants' integration, e.g., wages, health and subjective well-being of immigrants, relate to non-cognitive skills, which can easily be implemented using the same methodology offered in this paper.

## Supporting information

**S1 File.**
(PDF)

## Acknowledgments

We thank Olivier Bargain and seminar participants in Bordeaux University for their valuable comments and suggestions.

## Author Contributions

**Conceptualization:** Alpaslan Akay.

**Investigation:** Alpaslan Akay, Levent Yilmaz.

**Methodology:** Alpaslan Akay.

**Software:** Alpaslan Akay.

**Writing – original draft:** Alpaslan Akay, Levent Yilmaz.

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
