## [Decision Letter · Decision Letter 0]

1 Nov 2022

PONE-D-22-25219Non-Cognitive Skills and Labour Market Performance of ImmigrantsPLOS ONE

Dear Dr. Akay,

Thank you for submitting your manuscript to PLOS ONE. After careful consideration, we feel that it has merit but does not fully meet PLOS ONE’s publication criteria as it currently stands. Therefore, we invite you to submit a revised version of the manuscript that addresses the points raised during the review process.

We look forward to receiving your revised manuscript.

Kind regards,

Rong Zhu, Ph.D.

Academic Editor

PLOS ONE

Journal Requirements:

"No"

"There is no conflict of interest on the study."

7. PLOS requires an ORCID iD for the corresponding author in Editorial Manager on papers submitted after December 6th, 2016. Please ensure that you have an ORCID iD and that it is validated in Editorial Manager. To do this, go to ‘Update my Information’ (in the upper left-hand corner of the main menu), and click on the Fetch/Validate link next to the ORCID field. This will take you to the ORCID site and allow you to create a new iD or authenticate a pre-existing iD in Editorial Manager. Please see the following video for instructions on linking an ORCID iD to your Editorial Manager account: https://www.youtube.com/watch?v=_xcclfuvtxQ

8. Please include your full ethics statement in the ‘Methods’ section of your manuscript file. In your statement, please include the full name of the IRB or ethics committee who approved or waived your study, as well as whether or not you obtained informed written or verbal consent. If consent was waived for your study, please include this information in your statement as well.

Reviewers' comments:

Reviewer's Responses to Questions

**Comments to the Author**

1. Is the manuscript technically sound, and do the data support the conclusions?

Reviewer #1: Yes

2. Has the statistical analysis been performed appropriately and rigorously? 

Reviewer #1: Yes

3. Have the authors made all data underlying the findings in their manuscript fully available?

Reviewer #1: No

4. Is the manuscript presented in an intelligible fashion and written in standard English?

Reviewer #1: Yes

5. Review Comments to the Author

Reviewer #1: Comments:

(1) Abstract: "This paper investigates how non-cognitive skills, e.g., memory, empathy, attention, imagination, and social skills...". Memory is not one of the non-cognitive skills. It's usually considered as a part of cognitive ability (fluid intelligence, in particular). The paper does not use direct measures of "empathy, attention, imagination, and social skills". I suggest replacing part with Big-5 Personality traits.

(2) The paper examines the impact of non-cognitive skills on labor market performance. However, the only measure of the latter is employment. It is not clear why labor force participation and earnings are not examined.

(3) I am a bit concerned about the use of education as measuring your cognitive skills. Education is not only related to cognitive skills but also non-cognitive skills. As Heckman and Rubenstein (2001) and Cameron and Heckman (1993) indicate, level of education appears to be a mixed-signal. The authors may want to refer to other studies such as Agarwal and Mazumder (2013) regarding how cognitive skills are measured.

(4) This paper uses SOEP 1984-2013. I was wondering why more recent waves are not used.

(5) SOEP has other measures of non-cognitive skills such as locus of control. See, for example, Caliendo et al. (2022). The authors may want to consider this measure.

References

Agarwal, S. and B. Mazumder (2013). Cognitive abilities and household financial decision making. American Economic Journal: Applied Economics 5, 193–207.

Caliendo, M., Cobb-Clark, D.A.A, Obst, C., Seitz, H., and Uhlendorff, A. (2022). Locus of Control and Investment in Training. Journal of Human Resources, forthcoming.

Cameron, Stephen V., and Heckman, James J. 1993. The Nonequivalence of High School Equivalents. Journal of Labor Economics 11, 1-47.

Heckman, James J., and Rubinstein, Yona 2001. The Importance of Non-cognitive Skills: Lessons from the GED Testing Program. American Economic Review 91, 145-149.

6. PLOS authors have the option to publish the peer review history of their article (what does this mean?). If published, this will include your full peer review and any attached files.

Reviewer #1: No

---

## [Editor Report · Decision Letter 1]

16 Jan 2023

Non-Cognitive Skills and Labour Market Performance of Immigrants

PONE-D-22-25219R1

Dear Dr. Akay,

We’re pleased to inform you that your manuscript has been judged scientifically suitable for publication and will be formally accepted for publication once it meets all outstanding technical requirements.

Kind regards,

Rong Zhu, Ph.D.

Academic Editor

PLOS ONE

---

## [Editor Report · Acceptance letter]

15 Feb 2023

PONE-D-22-25219R1 

Non-Cognitive Skills and Labour Market Performance of Immigrants 

Dear Dr. Akay:

I'm pleased to inform you that your manuscript has been deemed suitable for publication in PLOS ONE. Congratulations! Your manuscript is now with our production department. 

Kind regards, 

on behalf of

Dr. Rong Zhu 

Academic Editor

PLOS ONE